# Discovery of Glucose Metabolism-Associated Genes in Neuropathic Pain: Insights from Bioinformatics

**DOI:** 10.3390/ijms252413503

**Published:** 2024-12-17

**Authors:** Ying Yu, Yan-Ting Cheung, Chi-Wai Cheung

**Affiliations:** 1Department of Anesthesiology, Laboratory and Clinical Research Institute for Pain, LKS Faculty of Medicine, The University of Hong Kong, Hong Kong SAR, China; u3007537@connect.hku.hk (Y.Y.);; 2Department of Anesthesiology, Queen Mary Hospital, The University of Hong Kong, Hong Kong SAR, China

**Keywords:** IVDD, hub genes, glucose, neuropathic pain

## Abstract

Metabolic dysfunction has been demonstrated to contribute to diabetic pain, pointing towards a potential correlation between glucose metabolism and pain. To investigate the relationship between altered glucose metabolism and neuropathic pain, we compared samples from healthy subjects with those from intervertebral disc degeneration (IVDD) patients, utilizing data from two public datasets. This led to the identification of 412 differentially expressed genes (DEG), of which 234 were upregulated and 178 were downregulated. Among these, three key genes (*Ins*, *Igfbp3*, *Plod2*) were found. *Kyoto Encyclopedia of Genes and Genomes* pathway analysis demonstrated the enrichment of hub genes in pathways such as the positive regulation of the ErbB signaling pathway, monocyte activation, and response to reactive oxygen species; thereby suggesting a potential correlation between these biological pathways and pain sensation. Further analysis identified three key genes (*Ins*, *Igfbp3*, and *Plod2*), which showed significant correlations with immune cell infiltration, suggesting their roles in modulating pain through immune response. To validate our findings, quantitative real-time polymerase chain reaction (qPCR) analysis confirmed the expression levels of these genes in a partial sciatic nerve ligation (PSNL) model, and immunofluorescence studies demonstrated increased immune cell infiltration at the injury site. Behavioral assessments further corroborated pain hypersensitivity in neuropathic pain (NP) models. Our study sheds light on the molecular mechanisms underlying NP and aids the identification of potential therapeutic targets for future drug development.

## 1. Introduction

Neuropathic pain (NP) is a pathological condition that arises from lesions or diseases of nerves, affecting both the peripheral and central nervous systems. Approximately 7% of the general population has been affected by NP [1], imposing a substantial personal burden on individuals, as well as tremendous economic impacts on societies. This burden is analogous to that of other public health issues, such as cancer, cardiovascular diseases, and obesity [2]. Clinically, NP is frequently a consequence of spinal cord injury (SCI) or peripheral nerve compression/injury [3], and patients usually present with allodynia and hyperalgesia [4]. In particular, intervertebral disc degeneration (IVDD), one of the most common clinical diseases worldwide, causes nerve compression [5], leading to NP. According to clinical data, 12%~35% of patients with disc degeneration suffer from NP [6]. Another meta-analysis has revealed that the prevalence of NP among patients with disc degenerative diseases was as high as 47% [7].

Pain is a complex phenomenon. In the initial exploration of the mechanisms underlying diabetic neuropathic pain, a potential connection between glucose metabolism dysregulation and pain mechanisms was hypothesized. Abnormal blood glucose levels disrupt the metabolism of neurons and glial cells, contributing to nerve damage and pain development [8,9]. This metabolic shift is not limited to DNP: evidence suggests the connections between metabolism irregularities and NP, including the derangement of the glycolytic pathway. Despite having a lower energy production efficiency than oxidative phosphorylation, glucose metabolism is preferred during the active phase of cell proliferation. Under normal conditions, pyruvate enters the tricarboxylic acid (TCA) cycle and undergoes oxidative phosphorylation, releasing CO_2_ and NADH. However, under hypoxic conditions, oxidative phosphorylation is either reduced or abolished, and pyruvate is converted into lactate and NAD^+^ via anaerobic glucose metabolism [10]. This holds true in the central nervous system, and the dysregulation of glucose metabolism is highly associated with pain. For instance, one report indicated that glycolytic shifts may be one of the initial changes in painful diabetic neuropathy; where such metabolic changes alter both the structures and functions of the dorsal root ganglion (DRG) and the nerve trunk, causing pain [11]. Other studies show implications of glial cells in NP [12]. Glial cells undergo a metabolic shift when activated by inflammation, transiting from oxidative phosphorylation to glucose metabolism as their major method of energy production. These switches, together with alterations in the pentose phosphate pathway and many others [13], not only provide substrates for various biosynthetic pathways but also regulate numerous intracellular signaling pathways at both transcriptional and post-transcriptional levels [14], amplifying their impacts on body functions.

The intricate interplay between the central nervous system and the immune system [15] is also key in the development of pain. Despite their distinct properties, numerous clinical and bench studies have demonstrated that neuroimmune interactions play a critical role in initiating and sustaining NP by affecting neural pain pathways [16]. During neuroinflammation, large amounts of inflammatory mediators, cytokines and chemokines are released, leading to behavioral hypersensitivity and allodynia [17]. In nervous system disorders associated with NP, functional dysregulations of pain-resolving immune cells and cytokine secretions have also been found [18].

Given the complex etiologies, pathophysiology and mechanisms that initiate and maintain NP, existing pharmacological treatments have been proven unsatisfactory. In hopes of finding better ways to alleviate NP, it is imperative to explore new molecules and mechanisms [19]. To date, NP animal models have predominantly focused on central nervous system mechanisms, whereas clinical research in humans has concentrated on peripheral blood biomarkers, creating a potential gap between systems. Studies investigating glucose metabolism and its impact on NP, particularly in the peripheral nervous system, remain limited. Although metabolic dysregulation has been implicated in cancer [20] and other CNS diseases [21], the specific role in the development and progression of NP induced by nerve injury is not well understood. Furthermore, while bioinformatics tools have been widely employed to identify metabolic genes, a systematic approach to investigate these genes in both clinical datasets and animal models has not been comprehensively undertaken in this field.

Our research thus represents a novel attempt to integrate bioinformatics analysis with experimental validation to elucidate the role of glucose metabolism in NP. By leveraging publicly available genomic datasets and advanced analytical methods, we aimed to identify metabolic genes linked to NP, followed by validation in a partial sciatic nerve ligation (PSNL) mouse model. This is, to our knowledge, the first attempt to integrate bioinformatics analysis of human genomic datasets with experimental validation in a rodent model to systematically explore the role of glucose metabolism-related genes in NP. With the bioinformatic methodologies and bench validations (RT-qPCR, and immunofluorescence), our work uniquely bridges clinical data analysis with preclinical validation in a PSNL mouse model. This integrative strategy provides novel insights into the molecular mechanisms of NP and lays the groundwork for potential metabolic-based therapeutic interventions.

## 2. Results

### 2.1. Identification of Differentially Expressed Genes in IVDD: 234 Upregulated and 178 Downregulated

Two datasets, GSE124272 and GSE150408, were retrieved from the GEO database for our analysis, comprising a total of 25 neuropathic pain (NP) samples and 25 healthy control samples. As these datasets originated from two separate projects, batch effect correction was necessary to ensure comparability. The effectiveness of batch effect mitigation is visualized in Figure 1A,B through principal component analysis (PCA) plots. Before correction, the datasets showed clear batch-related variability (Figure 1A), which was successfully reduced after correction (Figure 1B).

Following batch effect correction, PCA analysis confirmed improved integration between the two datasets, enabling reliable downstream analyses. A comparison of gene expression profiles between the NP and healthy control groups identified 412 differentially expressed genes (DEGs), including 234 upregulated and 178 downregulated genes. These results are visualized in a volcano plot, highlighting the overall distribution of DEGs (Figure 1C), and in a heatmap, which displays the top 20 upregulated and downregulated genes (Figure 1D).

### 2.2. Biological Processes, Cellular Components, and Molecular Functions Modulated in IVDD

To investigate the potential functions of all identified differentially expressed genes (DEGs), we performed gene ontology (GO) and *Kyoto Encyclopedia of Genes and Genomes* (KEGG) pathway analyses. GO analysis revealed that the upregulated DEGs were primarily enriched in biological processes related to the positive regulation of the ErbB and EGFR signaling pathways, as well as the response to reactive oxygen species (Figure 2A,B). For cellular components, significant enrichment was observed in tertiary granule lumens, specific granule lumens, and SCAR complexes, reflecting their involvement in immune and cellular dynamics (Figure 2C,D). Regarding molecular functions, DEGs were enriched in immune receptor activity, carbohydrate binding, and hormone activity, indicating their diverse functional roles (Figure 2E,F).

KEGG pathway analysis further highlighted the biological significance of these DEGs, showing enrichment in pathways such as transcriptional misregulation in cancer, NK cell-mediated cytotoxicity, and MAPK signaling (Figure 2G). These findings suggest that the identified DEGs may play key roles in cellular signaling and immune responses associated with IVDD.

To further explore the functional roles of differentially expressed genes (DEGs) in healthy individual and IVDD patients, gene set enrichment analysis (GSEA) and protein–protein interaction (PPI) network analysis were conducted. GSEA identified pathways enriched in both the healthy and disease (IVDD) groups. In the healthy group, key pathways included downregulated genes after 21 days vs. 1 day vaccination injection, genes upregulated in naive B cells activation vs. neutrophils, downregulated genes in stimulated neutrophils, upregulated genes in M-CSF-treated monocytes, and upregulated genes in CSF3-treated PBMCs. These results highlight distinct biological processes associated with normal physiological and steady immune functions in the healthy group (Figure 3A). In contrast, in the IVDD group, key pathways highlighted genes that were upregulated in myeloid cells compared to B cells and CD4^+^ T cells, as well as upregulated genes in lupus-associated myeloid cells compared to lupus-associated B cells and CD4^+^ T cells. Additionally, pathways enriched in myeloid cells compared to lupus samples suggest a shift toward myeloid-driven immune activity. These findings indicate a stronger innate immune response and suppressed adaptive immune responses in the diseased group (Figure 3B). PPI network analysis was conducted to explore the interactions among DEGs and to identify hub genes. The network revealed a tightly connected structure, with hub genes such as *Igfbp3* and *Ins* exhibiting high connectivity. This analysis highlights the central role of key DEGs in modulating the immune responses and metabolic-centered mechanisms underlying IVDD and potentially IVDD-induced neuropathic pain.

### 2.3. Identification, Expression Analysis, and Diagnostic Performance of Glucose-Related Genes in IVDD

To explore the role of glucose-related genes in IVDD, we identified the intersection of differentially expressed genes (DEGs) and glucose-related genes using a Venn diagram, yielding seven candidate genes for further analysis (Figure 4A). Two machine learning models, random forest (RF) and support vector machine (SVM), were applied to evaluate the diagnostic potential of these genes.

The RF model demonstrated superior performance, with lower residuals and a more concentrated residual distribution compared to the SVM model (Figure 4B,C). Receiver operating characteristic (ROC) curve analysis further confirmed the diagnostic accuracy of the RF model, achieving an AUC of 1.000, compared to 0.931 for the SVM model (Figure 4D). This indicates that the RF model is better suited for identifying disease characteristic genes.

Based on feature importance scores generated by the RF model, *Ins*, *Igfbp3*, and *Plod2* were identified as the most significant genes among the seven candidates (Figure 4E,F). These genes were selected for follow-up analyses due to their strong diagnostic potential for distinguishing between healthy and diseased groups.

We compared the expression levels of the three characteristic genes—*Ins*, *Igfbp3*, and *Plod2*—between the neuropathic pain (NP) and healthy groups. Both *Ins* and *Igfbp3* showed significantly higher expression in the NP group compared to the healthy group, whereas *Plod2* exhibited significantly lower expression in the NP group (Figure 5A–D). These results highlight distinct expression patterns of these genes in the diseased and healthy groups.

To further evaluate the diagnostic performance of these genes, receiver operating characteristic (ROC) curve analysis was conducted. The area under the curve (AUC) values for *Ins*, *Igfbp3*, and *Plod2* were 0.716, 0.689, and 0.713, respectively, demonstrating their moderate-to-strong ability to differentiate between healthy and NP groups. These findings underscore the potential of these genes as diagnostic biomarkers for neuropathic pain (Figure 5E).

### 2.4. Identification and Validation of Three Glucose-Related Genes with High Clinical Relevance and Predictive Accuracy for Diagnosing IVDD

To visualize the predictive ability of the identified characteristic genes, a nomogram was constructed using *Ins*, *Igfbp3*, and *Plod2* (Figure 6A). The nomogram integrates these genes into a scoring system, where the total score corresponds to a predicted probability of IVDD. This provides a practical tool for estimating disease risk based on gene expression profiles.

The clinical impact curve (Figure 6B) showed a close alignment between the predicted positive rate and the actual positive rate, demonstrating the reliability of the model in predicting IVDD. This indicates that the nomogram performs well in real-world applications, with minimal discrepancies between predictions and observations.

The decision curve analysis (DCA) (Figure 6C) further validated the clinical value of the nomogram. The DCA assesses the net benefit of using the nomogram across a range of threshold probabilities, revealing that the model offers significant advantages for clinical decision-making compared to default strategies such as treating all or no patients. These results highlight the utility of the nomogram in guiding clinical interventions.

Finally, the calibration curve (Figure 6D) demonstrated a strong agreement between the predicted and observed positive rates of IVDD diagnosis. The curve closely follows the diagonal reference line, confirming that the nomogram possesses robust diagnostic accuracy. Together, these analyses indicate that the nomogram is a reliable and clinically applicable tool for predicting the risk of IVDD, with potential for integration into future diagnostic workflows.

### 2.5. Immune Cell Infiltration in IVDD and Its Interaction with Glucose Metabolism-Associated Genes

A heatmap was drawn to illustrate the proportion of 22 types of immune infiltrating cells within the immune microenvironment of NP patients (Figure 7A). The correlations between immune infiltrating cells were further investigated in the NP group, and we confirmed the infiltration levels of each immune cell type in individual samples (Figure 7B,C). Twelve immune cell types, including activated CD4+ T cells, macrophages, mast cells, activated CD8+ T cells, and neutrophils, etc., were identified as differentially expressed between diseased and healthy groups (Figure 7D). We observed that the expressions of *Igfbp3* and *Plod2* were positively related to several immune cells, while *Ins* was negatively correlated with plasmacytoid dendritic cells and central memory CD8+ T cells (Figure 7E).

### 2.6. PSNL Model Successfully Changes Pain Behavior Without Inducing Motor Dysfunction

Behavioral tests were conducted to validate the PSNL model, and successful establishment of the pain model was confirmed (Figure 8A).

Mechanical hyperalgesia was continuously assessed using the von Frey test, starting from before surgery. Tests were halted during analgesic administration, then resumed 8 to 10 h after the final dose of analgesics on POD4. It was observed that the PWT in the PSNL group diminished from POD4 onwards, and persistently remained inferior to that of the sham group until the 14th day post-surgery (Figure 8B).

Thermal hyperalgesia was evaluated using the Hargreaves test. Compared to the sham group, the paw withdrawal latencies of the PSNL group were significantly lower on POD4, POD7, and POD14 (Figure 8C). On POD14, cold hyperalgesia was measured using the cold–hot plate method; however, no discrepancies were detected between the sham and PSNL groups (Figure 8D–F).

For gait analysis, we examined whether notable differences existed in any gait parameters between the two groups (Figure A1A,B in Appendix A). We found that the ipsilateral toe spread in the PSNL group was significantly narrower when compared to both its own contralateral toe spread and the sham group’s ipsilateral toe spread (Figure 8G).

### 2.7. NP Does Not Alter Blood Glucose Levels

On POD14, there was no significant difference in FBG between sham and PSNL groups. The glucose levels measured at 30-, 60-, 90-, and 120-min post-glucose loading also showed no significant difference between the two groups (*p* > 0.05, see Figure A1C in Appendix A).

### 2.8. IVDD-Altered Glucose Metabolism Gene Expression with Inflammation Elevated After PSNL

To investigate whether peripheral nerve ligation induces an immune response, qPCR was performed to quantify the levels of the pro-inflammatory factors *Tnfα* and *Il6*. Consistent with previous studies [22,23], *Tnfα* and *Il6* exhibited elevated levels on POD14 (*p* < 0.05); however, *Tgfβ* levels showed no significant difference between the PSNL group and the sham group (*p* > 0.05, Figure 9A). Besides, we validated the expression of specific genes using qPCR. On the 14th day post-PSNL, the mRNA levels of *Ins1* [24] and *Igfbp3* were drastically raised (*p* < 0.05, Figure 9B,D). However, the mRNA expression of *Plod2*, *Ins2,* and *Hif1α* (in blood and spinal cord samples) remained unaltered by PSNL (*p* > 0.05, Figure 9C,E, Figure A1D in Appendix A).

### 2.9. Increased Various Immune Infiltration near Injury Site in PSNL Model

Immune cell recruitment was observed in experimental groups. Glial cells, Schwann cells (SCs), and satellite glial cells, in particular, are located in peripheral nerves. Immunofluorescence staining analysis revealed that in normal sciatic nerves, calcium-binding protein S100β was primarily localized within the myelin sheath. The level of S100β in the PSNL model group exhibited a significant increase compared to the sham operation group (*p* < 0.05), and notably gathered around the injury site of the sciatic nerve (Figure 10A–C, Figure A1F in Appendix A). Furthermore, macrophage marker F4/80 and M1 marker CD86 co-staining demonstrated higher levels in the PSNL group compared to the sham group (*p* < 0.05, Figure 10D–F), with F4/80 positive cells clustering near the injury site (Figure A1E in Appendix A). CD11c^+^ F4/80^neg^ cells, identified as dendritic cells within the peripheral nervous system by previous studies [25,26], presented a higher level in the PSNL group (*p* < 0.05, Figure 10G–I). However, Cd16^+^—the most potent activating receptor on NK cells [27]—showed no difference in numbers between sham and PSNL groups (*p* > 0.05, Figure A1G–I in Appendix A).

## 3. Discussion

In this study, we utilized bioinformatic analyses to identify glucose metabolism-related genes in NP and validated our findings in the PSNL animal model. Our primary findings include that (i) three glucose metabolism-related genes—*Ins*, *Plod2* and *Igfbp3*—potentially alter the risk of NP in IVDD samples as compared to healthy ones. (ii) Macrophages, SCs, and dendritic cells showed increased infiltration levels in high-risk NP patients when compared to low-risk individuals; (iii) the relationship of *Ins1* and *Igfbp3* expressions with NP was confirmed in the animal PSNL model; and (iv) immune cell infiltration was found to be abundant around the site of sciatic nerve ligation in the PSNL model.

In the study of connections between glucose metabolism and NP, how specific genes would affect NP has not been fully elucidated. For the first time, glucose metabolism-related genes affecting NP in IVDD patients have been identified, with the help of bioinformatic analyses, a nomogram, and validations via a NP mice model.

### 3.1. Ins1/2

In rodents, two non-allelic insulin genes—*Ins1* and *Ins2*—are present, both enriched in pancreatic β-cells [28]. *Ins1* is a retrotransposon specific to rodents, whereas *Ins2* is an ortholog to the insulin genes found in other mammalian species [29]. In mammals, beyond the endocrine pancreas β-cells, INS productions has been observed in neurons, the brain [30], and other tissues [31]. INS binds to insulin receptors (IRS), triggering a cascade of events and intracellular signaling networks. In particular, the PI3K pathway regulates metabolic responses and cell growth via Akt. Akt-mediated metabolism is stimulated by the activation of enzymes involved in gluconeogenesis, glucose uptake, protein synthesis, and adipogenesis, while cell growth response is primarily induced by the mTOR pathway [32]. In dysregulated states such as type I diabetes, INS signaling is impaired, leading to disrupted glucose metabolism and glycogenesis [33].

Glucose metabolism regulated by INS is associated with pain. Previous studies discovered that the activation of *Ins1* receptors can elicit an upregulation of glycolytic activity, thereby promoting a reduction in insulin resistance through macrophage polarization and enhancement of energy expenditure [34]. However, raised energy demand in tissues would enhance their dependence on aerobic glucose metabolism, leading to increased extrusion of metabolites (protons and lactate), and hence the sensitization of primary afferents. With the reprogramming of sensory neuron metabolism and aerobic glucose metabolism, pain ensues [35]. Furthermore, glycolysis, as a key component of glucose metabolism, may exacerbate neuropathic pain [36].

In our study, we demonstrated an increase of *Ins1* mRNA expression on POD14 via qPCR while, on the same day, the von Frey test showed a decreased PWT. Although no changes in *Ins2* levels were observed in the PSNL group, the coexistence of elevated *Ins1* mRNA expression and increased mechanical withdrawal threshold could point to correlations between the two. One explanation for this could be that insulin reduces mechanical stimuli-induced currents in sensory neurons to lower activation thresholds [37]. For instance, previous studies have proved that most sensory neurons with IRS co-express transient receptor potential channel subfamily vanilloid member 1 (TRPV1). This has two implications: one, it is likely that insulin binding to neuronal IRS produces intermediate molecules which sensitize and/or activate nociceptive channels, such as TRPV1; two, by decreasing the threshold of TRPV1 activation, insulin increases TRPV1 sensitivity. Since TRPV1 can be activated by different types of nociceptive stimuli—no matter whether exogenous, endogenous, physical, or chemical—it is likely that the sensitization of TRPV1 by insulin causes more nociceptive signals to be integrated and potentiated, causing pain [38].

On the other hand, unvaried *Ins2* levels might be explained by the negligible difference in glucose tolerance between the sham and PSNL groups, and this is consistent with previous research [39]. Despite their close relationship, *Ins1* and *Ins2* have different promoter elements, tissue- and temporal-specific expression patterns, and imprinting status [40]; thus, we hypothesize that these differences cause functional variations, such that *Ins1* and *Ins2* levels may not change in the same direction. Such a concept is echoed by N. Babaya et al. [41], who demonstrated the functional deficit of the *Ins1* gene relative to the *Ins2* gene through diabetic studies. Additionally, it was observed that *Ins2*^−/−^ mutants did not present with major metabolic disorders [42]. Past reports have also shown that during hypoglycemia, glucose concentration in the extracellular fluid of the brain is reduced, and the amount of *Ins2* mRNA detected per neurogliaform cell is decreased [43]. At constant glucose concentrations, it could be expected that *Ins2* mRNA levels would carry little to no variations. These, together with our findings in the OGTT test, might explain the unvaried level of *Ins*2 observed in this study.

### 3.2. Igfbp3

The IGF axis comprises various components, including ligands, such as insulin (INS) and insulin-like growth factors 1 and 2 (IGF1, IGF2); receptors, such as IGF1R, and IGF2R; as well as IGF binding proteins (IGFBPs) 1 to 7 [44]. Among these, IGFBPs have particularly high affinities for IGFs and are known to regulate the bioactivity of IGFs both positively and negatively.

As the most abundant subtype of IGFBP [45], IGFBP3 is the main carrier of IGF-1 in plasma [46]. Through the binding and releasing of IGF, IGFBP3 regulates the access of IGF to IGF1R, modulating its biological activities. This includes increasing metabolism and cell survival, as well as activating glucose metabolism by phosphorylation of AKT through IGF1R signaling. This mechanism has been used in therapeutic drugs, such as the glycolytic inhibitor 2-deoxyglucose (2-DG), which could disrupt the interaction between IGF-1 and IGFBP3 [39].

Given its importance, the role of IGFBP3 in glucose metabolism has also been investigated in multiple studies, where *Igfbp3* expression was reported to be related to insulin levels in obese patients undergoing sleeve gastrectomy and IGF1 levels in rehabilitating non-small cell lung cancer (NSCLC) patients with exercise interventions [47,48].

Using qPCR, we found that *Igfbp3* expression levels were increased in both IVDD patients and PSNL mice. Clinically, the elevation of *Igfbp3* has been implicated in a number of conditions. In one study, comparison of serum in clinically active rheumatoid arthritis (RA) patients and 51 control subjects showed elevated *Igfbp3* in RA patients [49]. Another study discovered that the increase in tear *Igfbp3* levels in type 2 diabetes mellitus (T2DM) patients correlated with damage to the corneal subbasal nerve plexus [50]. In the peripheral blood of patients with painful bone metastases, raised *Igfbp3* expression was found [51].

Besides chronic low-grade inflammation and inflammation-related pathways, *Igfbp3* is also positively related to immune cell infiltration. For instance, in hypertrophic cardiomyopathy patients with high *Igfbp3* expression, immune cell infiltration was more enriched [47,48]. Similarly, we found that IGFBP3was enriched in T cells and myeloid-derived suppressor cells (MDSC) in NP patients, suggesting a potential involvement of IGFBP3 in modulating immune responses.

### 3.3. Plod2

PLOD2, a member of the PLOD family (PLOD1, PLOD2, and PLOD3), plays a key role in glucose metabolism and the maintenance of collagen stability [52]. Primarily initiating the hydroxylation of lysine in collagen [53], PLOD2 creates attachment sites for carbohydrate molecules, which are crucial for the stability of collagen cross-links. This function highlights PLOD2’s importance in various physiological and pathological conditions.

There have been some descriptions of the involvement of *Plod2* in different conditions. In Bruck syndrome, mutations in *Plod2* were shown to be causative of collagen I underhydroxylation, leading to osteoporosis, scoliosis, and joint contractures [54]. On the other hand, in the SCI model, *Plod2* upregulation was proven to be able to promote spinal neuron regeneration after injury, potentially relating to the modification and reorganization of collagen fibers [55]. Furthermore, *Plod2* has been reported to exhibit a high expression in tumors. Due to an inappropriate regulation of the plods family in tumors, PLOD2 triggers extracellular matrix (ECM) collagen remodeling in the tumor microenvironment (TME), and as a result of increased collagen content in solid tumors, tumor infiltration, invasion, and distant metastasis are fostered [56].

Recent studies have suggested that PLOD2 may play a role in glucose metabolism by promoting aerobic glycolysis through the upregulation of hexokinase 2 (HK2) [57]. This interaction is thought to support tumor cell proliferation and invasion by increasing glucose uptake and lactate production.

In our bioinformatics study, we found that the expression level of *Plod2* was much higher in healthy subjects than that of diseased persons. However, it was found that *Plod2* levels remained the same in both PSNL-treated mice and sham mice in our study, suggesting that PLOD2’s role in glucose metabolism might be context-dependent. We hypothesize that such discrepancies are caused by the difference in oxygen supply between PSNL and IVDD. In fact, different types of NP have different pathophysiologies. Although hypoxia has been observed in the DRGs and dorsal horn of the spinal cord in diabetic NP [58], NP in PSNL has shown greater association with inflammatory pain [59]. The microenvironment of IVDD was also discovered to be hypoxic [60]. Under normal oxygen supply conditions, *Plod2* promotes tissue and ECM regeneration; and in IVDD individuals, its levels are decreased. This might, in part, explain why bioinformatic analysis results could not be replicated in animal models.

Subsequently, we turned to investigating the relationship between *Plod2* and hypoxia. In particular, we looked into hypoxia-inducible factor (HIF)-1—which plays an important role in bodily responses to hypoxia [61]—and found that acute hypoxia demonstrably triggered *Hif1α* activation in vitro, while chronic hypoxia was marked by a significant loss of *Hif1α* [62]. PLOD2 protein could bind to HIF1 Protein, activating the PI3K-AKT pathway. This has been proven to promote tumor cell infiltration and proliferation in the tumor microenvironment [63]. Nonetheless, in an attempt to quantify and compare *Hif1α* expression levels in the spinal cord and blood samples of both sham and PSNL groups, no substantial variance in *Hif1α* expression was found (see Figure A1D in Appendix A). Hence, further studies might be needed to establish whether a hypoxic microenvironment is present in NP.

### 3.4. Immune Cell Infiltration

Numerous studies have investigated the metabolic pathways utilized by immune cells during their inflammation and activation. In particular, macrophages/microglia and astrocytes were found to employ different metabolic pathways that are precisely controlled by various metabolic regulators, such as the mechanistic target of rapamycin (mTOR), glucose transporter, etc. Upon activation, cells undergo metabolic changes to fulfill the demands for their phenotypic switch [64]. The onset of inflammatory responses has also been shown to be intricately linked with the emergence of insulin resistance, with studies showing that insulin administration can have positive effects on memory and cognitive functions in animal models [65].

Through ssGSEA analyses, we identified the types of immune cells infiltrated—including monocytes/macrophages and dendritic cells—into IVDD patients’ peripheral blood. Similarly, via immunofluorescence, we observed an increased infiltration of glia cells, macrophages, and dendritic cells into the sciatic nerve of PSNL mice on POD14, on the date where behavioral tests presented dramatic changes. The increased infiltration was particularly apparent at the distal extremity of the ligation site.

The peripheral nervous system lacks lymphatic vessels and is protected by the blood–nerve barrier (BNB), which prevents lymphocytes and antibodies from entering the neural parenchyma [66]. However, as the peripheral nervous system sustains damage/injury, the integrity of the BNB is destroyed, and an intricate cellular immune response ensues [67]. After injury, neutrophils aggregate at the distal stump [68]. Two days after injury, as part of the Wallerian degeneration process, SCs de-differentiate and actively initiate the fragmentation, disintegration, and phagocytosis of axons and myelin [69]. Within 3~5 days after injury, surrounding glial cells, SCs, dendritic cells, and macrophages engage and clear the resulting debris. This process potentially facilitates the successful regeneration of the injured proximal nerve fiber [70]. As early as 5 days post nerve repair, regenerating axons “stagger” across the suture site, intermingling with SCs. These SCs extend along the basal lamina of the denervated endoneurial tubes to guide, support, and direct outgrowing axons towards their innervation sites [71].

Immune cell infiltration has always been a field of interest. Previously, a study investigating SCs in the chronic constriction injury (CCI) model showed a decline in the number of SCs and a separation of SCs from the axons of the sciatic nerve on POD21. This phenomenon was attributed to SC pyroptosis [72]. Nonetheless, we found an increase in the number of SCs on POD14, which might suggest a unique immune cell infiltration pattern in the PSNL model. While the CCI model represents an injury paradigm where major axonal degeneration is delayed due to secondary events [73,74], no delayed WD has been reported in the PSNL model. Hence, we hypothesize that the transformation of SCs into their regenerative phenotype following CCI had not yet begun at the time of sampling (POD21) in the mentioned study, due to delayed WD. Besides, the marker S100 is not expressed on regrowing SCs at the axotomized site [75], which might mean that the cells stained by S100 in immunofluorescence are not representative of the whole SC population.

### 3.5. Novelty and Limitations

The novelty of our study lies in its exploration of the relationship between glucose metabolism, peripheral immune infiltration, and neuropathic pain (NP), bridging clinical observations with bench research. Firstly, the relationship between glucose metabolism and peripheral neuropathic pain was initially discovered in the context of diabetic peripheral neuropathy (DPN) [76]. Extensive research has shown that impaired glucose regulation plays a critical role in DPN pathogenesis [77]. Notably, metformin—a widely used antidiabetic drug—has been shown to alleviate DPN by improving glycemic control, which aligns with its primary therapeutic mechanism [78]. However, findings from studies on DPN and metabolic regulation cannot be fully extrapolated to NP as a broader category. This is because diabetes, particularly type 1 diabetes, involves complex pathological mechanisms [79,80], and analyses of type 2 diabetes are often confounded by factors such as dietary habits, age, and socioeconomic parameters (SEP) [81,82]. Thus, there remains a gap in understanding how glucose metabolism independently contributes to NP, beyond diabetic conditions. Our study seeks to address this gap by focusing on NP induced by nerve injury, independent of the systemic complications associated with diabetes. Additionally, sedentary lifestyles and lack of exercise have contributed to a rising prevalence of intervertebral disc degeneration (IVDD) and related conditions, such as sciatic nerve compression [83]. This highlights the relevance of choosing IVDD datasets for our research, as they represent a population commonly afflicted by neuropathic pain. Notably, many patients with sciatica or sciatic nerve pain do not exhibit impaired glucose levels [84], a finding that aligns with observations from our mouse models. This consistency further underscores the importance of focusing on glucose metabolism in NP, independent of systemic glucose dysregulation. Secondly, this study fills the research gap on glucose metabolism and immune infiltration in NP: NP is notoriously difficult to treat due to its complex and poorly understood mechanisms. Existing research predominantly emphasizes the role of neuro–immune interactions in the development of NP, highlighting the impact of inflammatory mediators and immune cell activity on the central nervous system. However, there has been limited investigation into the interplay between glucose metabolism and peripheral immune infiltration in NP. Our study fills this gap by systematically examining the association between glucose metabolism-related genes and peripheral immune responses in NP. By integrating bioinformatics analysis of clinical datasets with controlled bench experiments, we provide novel insights into how glucose metabolic shifts and immune cell dynamics contribute to the maintenance of NP. Specifically, GSEA results highlight distinct biological processes associated with steady-state immune functions in the healthy group. The low expression of genes downregulated 21 days after vaccination compared to the acute phase suggests a baseline immune state. Additionally, the reduced expression of genes upregulated in M-CSF-treated monocytes indicates that monocytes remain in a resting state. These findings collectively support the notion that the healthy group represents a physiological baseline characterized by the absence of significant immune activation. By systematically validating gene expression patterns in the healthy group, this study establishes a robust baseline for comparison. This innovative approach provides novel insights into baseline physiological conditions, often underexplored in neuropathic pain research. The observed patterns not only highlight the utility of GSEA in characterizing immune system dynamics under normal conditions but also serve as a critical reference for identifying deviations in diseased groups. Lastly, bridging clinical and bench research: one of the most significant contributions of our work is the successful integration of clinical observations with bench research. Clinical studies leveraging large-scale genomic data provide valuable macro-level insights, but they often suffer from confounding variables and a lack of control over experimental conditions. Conversely, bench studies excel at isolating specific variables under controlled environments, but they can struggle to translate findings into real-world clinical applications. Our study addresses this disconnect by using bioinformatics tools to identify glucose metabolism-related genes associated with NP in human datasets, followed by validation in a partial sciatic nerve ligation (PSNL) mouse model. This dual approach ensures that our findings are both clinically relevant and mechanistically robust, facilitating the translation of bench research into potential clinical applications. Importantly, we are currently conducting ongoing studies to further investigate these genes in the peripheral nervous system, providing a foundation for future peripheral-targeted therapeutic interventions.

This study may have some flaws. Firstly, data from IVDD patients were used for analysis. IVDD patients are usually accompanied by sciatic nerve compression and/or lower-back pain, but IVDD also carries other pathological changes such as apoptosis of the fibrous annulus. Whether the PSNL model can completely simulate the neuralgia caused by IVDD remains uncertain, which could also explain the inconsistency between the gene expression changes in the PSNL model and IVDD patients. Future studies could explore alternative or complementary animal models that better simulate the multifaceted pathology of IVDD. Additionally, integrating ex vivo models derived from human IVDD tissues might also provide a more accurate representation of disease-specific mechanisms. Secondly, bioinformatics results made predictions of gene enrichment and different immune infiltration cell types, but we did not demonstrate the co-expression of related genes and specific immune cells, which may require investigation by subsequent studies. To resolve this, single-cell RNA sequencing (scRNA-seq) combined with spatial transcriptomics could be employed in future studies to precisely map gene expression to specific cell populations co-related with glucose metabolism and immune responses in both clinical and preclinical samples. Thirdly, we validated the gene enrichment findings from bioinformatics analyses via a NP mouse model, but such findings may require more validations in human samples later. To achieve this, prospective studies could involve collecting peripheral blood or nerve biopsy samples from patients with NP, followed by RNA sequencing or immunohistochemical analyses. Finally, we have validated the expression of the identified genes in the whole blood of NP mice, demonstrating good consistency with our bioinformatics predictions. However, considering that clinical interventions often focus on peripheral regions, we are currently extending our research to investigate these genes directly within the sciatic nerve tissue. Specifically, we are utilizing immunofluorescence techniques to observe co-staining specific gene discovered in this study and immune cell infiltration of the injured sciatic nerve, providing a basis for sciatic nerve interventions. By targeting the peripheral sciatic nerve, we hope to develop more precise therapeutic approaches tailored to the peripheral injury site of NP. These studies are actively underway and represent a key direction of our research.

## 4. Materials and Methods

### 4.1. Data Acquisition

Gene expression profiling data were first downloaded from the GEO database, which contains a large amount of microarray data submitted by research institutions. Then, the GSE124272 and GSE150408 datasets, which contain data from patients with IVDD-induced NP, were selected for subsequent analysis. Genetic data of 8 IVDD patients with NP and 8 healthy controls were obtained from the GSE124272 dataset (GPL21185 Agilent-072363 SurePrint G3 Human GE v3 8x60K Microarray), whereas the GSE150408 dataset (GPL21185 Agilent-072363 SurePrint G3 Human GE v3 8x60K Microarray) provided genetic data of 17 NP patients and 17 healthy controls. These 2 datasets were merged, and the gene expression data were annotated and normalized by Perl script.

### 4.2. Differentially Expressed Genes Analysis

Preliminary selection of relevant genes was carried out by DEGs analysis, using the limma package [85]. The preliminary selection of relevant genes was carried out by differential expression analysis using the limma package (version 3.52.1), a robust tool for RNA-sequencing and microarray studies. The identified DEGs were made sure to meet the following criteria: *p* < 0.05 and |log2 fold change (FC)|  > 0.5. Then, the volcano map and heatmap derived from the ggplot2 package (Version 3.4.1) were used to present the DEGs.

### 4.3. Functional Enrichment Analysis

After obtaining the desired DEGs, we explored their possible effects by functional enrichment analyses. Gene Ontology (GO) and *Kyoto Encyclopedia of Genes and Genomes* (KEGG) were used with the org.Hs.eg.db (Version 3.16.0) and clusterProfiler packages (Version 4.6.0) [86,87,88]. A *p* < 0.05 was considered significant. Then, gene set enrichment analysis (GSEA) was performed in the GSEA 4.2.0 software, and immune-related datasets were regarded as reference gene sets [89]. A false discovery rate (FDR)  <  0.05 and *p*  <  0.05 were set as selection criteria. Finally, the search tool in the Retrieval of Interacting Genes (STRING; https://string-db.org) database was used to develop a protein–protein interaction (PPI) network. Nodes represented the proteins, and the line thicknesses showed the strength of the interactions [90].

### 4.4. Identification of Target Gene Biomarkers

Using the Venn Diagram Generator, a Venn diagram was created to identify the overlap between the DEGs and glucose metabolism-related genes. The Venn diagram illustrating the overlap between differential expression genes (DEGs) and glycolytic genes was generated using R software (version 4.3.0) with the VennDiagram package (Version 1.7.3). The code used was adapted from the publicly available example provided by the R Graph Gallery (https://r-graph-gallery.com/14-venn-diagramm, accessed on 12 July 2023). Modifications were made to adjust the diagram’s color scheme, transparency, and layout to meet the specific requirements of this study. Various methods could be used to further select target genes, including the random forest (RF) [91] and support vector machine (SVM) [92] techniques. RF is an algorithm that integrates multiple trees through ensemble learning, whereas SVM is a flexible supervised machine learning algorithm widely used in statistical classification and regression analysis. In order to find out whether the RF or SVM technique would provide more accurate and reliable results, receiver operating characteristic (ROC) curves were drawn to visualize and compare their marker screening abilities. After target genes were chosen, their gene expressions in both disease and healthy control groups were displayed using boxplots and heatmaps.

### 4.5. Construction of a Nomogram

Based on the target genes identified by machine learning, we constructed a nomogram with the “foreign” package (version_0.8-84). A clinical impact curve was generated to assess the difference between the predicted positive rate and the true positive rate of the nomogram, while a calibration curve was drawn to present the accuracy of the nomogram. Using the rmda package (version 1.6), a decision curve analysis (DCA) was also constructed to evaluate the clinical application value of the nomogram [93].

### 4.6. Immune Cell Infiltration Analysis

Single-sample gene set enrichment analysis (ssGSEA) is commonly used in immune cell infiltration analysis. The principle of this method is to compare the gene expression data of each sample with a specific gene set to estimate the relative enrichment. In this study, we used the ssGSEA in the GSVA package to investigate the differences of immune cell infiltration between disease samples and controls [94]. Then, the infiltration levels of 28 immune cell types were presented in heatmap and violin plots. Furthermore, we conducted Spearman correlation analysis on the expression level of target genes and various immune cells.

### 4.7. Animals

Adult male C57BL/6J mice aged 8~10 weeks were procured from the Specific Pathogen Free Breeding Area of the Centre for Comparative Medicine Research (CCMR) at Li Ka Shing Faculty of Medicine. The mice were housed in accordance with a 12 h light cycle/dark cycle (lights on at around 07:00; lights off at around 19:00) and provided with normal chow (Lab Diet 5012) and water ad libitum. All experiments and procedures were conducted in compliance with the regulations set forth by the Committee on the Use of Live Animals in Teaching and Research (CULATR No. 5452-20) at the University of Hong Kong. Animal care and housing were implemented in accordance with the guidelines established by CCMR. Unless part of time course studies, the animals were subjected to sacrifice on day 14 following PSNL.

### 4.8. Model Establishment

Partial sciatic nerve ligation (PSNL) is a well-established procedure widely used to study NP. In our study, PSNL surgery was performed according to the methodology established by Seltzer et al. [95]. Mice were administered with 5% isoflurane for the induction of anesthesia and 2.5~3% for maintenance. After making sure the mice were fully anesthetized, PSNL surgery was initiated by making an incision in the skin, and the muscle was passively separated at the left mid-thigh level to expose the left sciatic nerve. Then, a tight ligation using a 7# silk thread was tied around approximately 1/3~1/2 diameter of the sciatic nerve. Following the procedure, the incision wound was closed by a 5# cotton thread. In the sham group, all the procedures mentioned above were performed, but without any manipulation of the sciatic nerve.

### 4.9. Fasting Blood Glucose Test and Oral Glucose Tolerance Test

The fasting blood glucose (FBG) and oral glucose tolerance test (OGTT) were conducted based on prior studies [96,97]. Mice had their blood glucose levels tested if they had either fasted for 6 h or had been re-fed (within 0 min to 2 h of re-feeding). Re-feeding was performed using a glucose bolus dosage (1 g/kg of body weight) administered through gavage. The initial blood droplet was discarded, and the second droplet was favored for assessment with a glucometer (Roche Diagnostics, Mannheim, Germany). Blood samples from the tail were obtained at 0, 15, 30, 60, and 120 min after glucose ingestion.

### 4.10. Mechanical Allodynia Assessment: Von Frey Test

In line with Gautam’s protocol [98], mice were placed individually in plastic enclosures on a stainless-steel rack with meshed flooring for roughly 60 min to allow for habituation. An electronic von Frey filament aesthesiometer (IITC Life Science Inc, Woodland Hills, CA, USA) was used to automatically record the paw withdrawal threshold (PWT), which indicates the force experienced by the bent von Frey filaments when a positive response is evoked. The filaments were slowly and perpendicularly applied to the plantar surface of the left hind paw, proximate to the heel. Quick withdrawal or paw licking were characterized as positive reactions, while responses related to basic locomotion were disregarded. Five measurements were taken at intervals of 5 min for each animal, and the mean value of the five measurements, excluding the highest and lowest readings, was calculated as the PWT for that animal.

### 4.11. Thermal Allodynia Assessment: Hargreaves Test

The Hargreaves test was conducted following a similar methodology to those used in previous studies [99]. For the assessment of heat hyperalgesia, the animals were individually placed in small plastic chambers positioned on a smooth glass surface. At least 60 min were allotted for acclimatization prior to the test. Utilizing a paw analgesia meter (IITC Life Science Inc, Woodland Hills, CA, USA), a focused source of radiant heat was projected onto the plantar surface of the left hind paw from beneath the glass surface. The time taken for the mouse to withdraw its hind paw was measured. To prevent harm to the paw, the maximum cutoff time for heat projection was set to 20 s, after which the stimulus would be withdrawn regardless of response. A minimum interval of 5 min was maintained between each measurement. In total, 5 measurements were recorded and the average of the 5 measurements was calculated to determine the paw withdrawal latency.

### 4.12. Motor and Sensory Assessment: Low-Cost Gait Analysis

Gait analysis was conducted following the methodology outlined by Virginia Wertman et al. [100]. The dimensions of the testing tunnel were customized to be 2.5 inches in width, 3 inches in height, and 13 inches in length. The goal chamber utilized for the experiment was opaque and painted in a dark color. To ensure proper acclimatization, mice were placed in the new environment for 30 min before behavioral assessment. Furthermore, given the nocturnal nature of mice, all subjects were made sure to be fully awake and alert for a minimum of 5 min prior to testing. To track the mice’s movements, non-toxic, washable, water-based paint was utilized. Two contrasting colors, namely blue and red, were assigned to the hind limbs and forelimbs of all toes respectively. A mouse was positioned at the starting point of the tunnel and allowed to traverse the entire length until reaching the goal chamber. In order to assess the gait accurately, 4 to 6 steps with consistent spacing marked by clear, non-smudged footprints were employed for scoring. Information like stride length, stride width, and toe spread were recorded for subsequent analysis.

### 4.13. Whole Blood Collection and Preprocessing

Mice were euthanized with pentobarbital sodium (200 mg/kg) on postoperative day 14 (POD14). Using 1 mL syringes, blood samples were directly collected from the heart of each animal, and the gathered volume represented the whole blood. Samples were placed in a 50 mL polypropylene conical centrifuge tube and mixed with a volume of RBC lysis buffer (eBioscience™, San Diego, CA, USA) 10 times that of the collected blood. The mixture was left standing at room temperature for 10 min, after which cell pellets were isolated through centrifugation at 600× *g* (1400 rpm) for 10 min at room temperature. Subsequently, the supernatant was cautiously decanted, and the pellet was gently resuspended in 500 µL of RBC lysis buffer before being transferred to a 1.5 mL microtube (Corning, Corning, NY, USA). The blood sample was set aside for 5 min and was centrifuged again in a microfuge at room temperature, at a speed of 3000 rpm for 2 min. Afterwards, the supernatant was cautiously aspirated, and the pellet was resuspended in 500 ul of sterile DPBS. Finally, the cells were pelleted one more time by spinning them in a microfuge at room temperature, running at 3000 rpm for 2 min. The supernatant was then precisely aspirated, leaving the desired cellular component for RNA extraction. Such extraction procedures will be described in detail in a subsequent section.

### 4.14. RNA Extraction (Whole Blood Samples)

RNA extraction followed Kang JE’s protocol [101]. Total RNA was isolated from both the cellular component of the whole blood sample using Trizol (Takara, Tokyo, Japan), chloroform (Sigma-Aldrich, Burghausen, Germany), and isopropanol (Sigma-Aldrich, Burghausen, Germany) according to the manufacturer’s instructions. Extracted RNA was quantified by measuring its absorbance at 260 nm and 280 nm by EPOCH 2 microplate reader and the BioTek Gen5 software (version 2.0, Agilent BioTek, Santa Clara, CA, USA).

### 4.15. Real-Time Quantitative Polymerase Chain Reaction (qPCR)

Reverse transcription (RT) from RNA to cDNA was performed with PrimeScript RT Master Mix (Takara, Tokyo, Japan). According to the manufacturer’s instructions, RT was performed at 37 °C for 30 min and 85 °C for 5 s. cDNA samples were diluted and stored at −20 °C until further analysis. qPCR was performed on a StepOnePlus™ Real-Time PCR System (Applied Biosystems, Waltham, MA, USA) or Roche LightCycler 480 (Sigma-Aldrich, Burghausen, Germany) using TB Green intercalation dye (Takara, Tokyo, Japan) according to manufacturer instructions. Primers of *Ins1*, *Ins2*, *Igfbp3*, *Plod2*, *Hif1*, *Il6*, *Tnfα*, *Tgfβ*, and internal reference gene β-actin were manufactured by Integrated DNA Technologies (IDT, Singapore) and listed in Table 1. Relative gene expression was measured by the conventional 2^−ΔΔCT^ method and normalized to the internal reference gene β-actin.

### 4.16. Immunofluorescence

Mice were sacrificed with pentobarbital sodium (200 mg/kg) and perfused from the left ventricle with ice-cold 0.1 M phosphate buffer (PB), followed by 4% ice-cold paraformaldehyde. The left sciatic nerve was harvested up from the sciatic notch to the nerve branch, including the ligation site. The tissue was then postfixed with paraformaldehyde for 4~6 h and dehydrated by 15% and 30% sucrose solutions sequentially. After embedding in Tissue-Tek O.C.T Compound (Sakura Finetek, Torrance, CA, USA), 30 μm sections were cut such that the nerve stem and its roots could be visualized in the same plane (Leica 3050S) (Leica, Wetzlar, Germany). The floating section method was employed. Antigen retrieval was performed using pH 8.4 0.01 M sodium citrate (Powder: Sigma-Aldrich, Burghausen, Germany) solution for 30 min in a water bath set to 80 °C. The samples were allowed to cool at room temperature for 10~15 min after antigen retrieval and washed by 0.1 M PB 3 times at 5-min intervals.

Sections were blocked using a solution of 10% donkey serum (Sigma-Aldrich, Burghausen, Germany), 1% BSA (Rockland, Washington, DC, USA), and 0.1% Triton-X (Sigma-Aldrich, Burghausen, Germany) at room temperature for 3 h. They were then washed in 0.1 M PB for 10 min, and subsequently incubated with rabbit anti-F4/80 antibodies (1:80; Abcam, Cambridge, UK), mice anti-Cd11c antibodies (1:80; Abcam, Cambridge, UK), mice anti-Cd86 antibodies (1:100; Dako, Carpinteria, CA, USA), and mice anti-S100β antibodies (1:100; MilliporeSigma, Darmstadt, Germany) at 4 °C for 2 nights. After primary antibody incubation, the sections were again washed in 0.1 M PB. To prepare for microscopic visualization, slices were further incubated with fluorescent-conjugated secondary antibodies (1:500, donkey anti-rabbit Alexa-568; Invitrogen, Waltham, MA, USA and 1:500, donkey anti-mice Alexa-488; Invitrogen, Waltham, MA, USA), washed in 0.1 M PB, and stained with Dapi (1:2000; Abcam, Cambridge, UK) for 20 min. Finally, the slides were mounted with mounting medium with anti-fade properties. The images were captured by Zeiss LSM 900 confocal microscope (Zeiss, Jena, Germany).

### 4.17. Statistical Analysis

All data are presented as mean ± standard error of the mean (SEM) and were analyzed using GraphPad Prism version 10.0 (GraphPad, San Diego, CA, USA). Statistical comparisons between two groups were conducted using an unpaired two-tailed Student’s *t* test if the data were normally distributed and exhibited homogeneity of variances; otherwise, a Mann–Whitney U test was applied. For multiple group comparisons, data were analyzed using one-way ANOVA followed by Tukey’s multiple comparisons test. If data did not meet the assumptions for parametric analysis, a Kruskal–Wallis test with Dunn’s post hoc test was used. In all analyses, *p* < 0.05 was considered statistically significant.

### 4.18. Tools and Visualization

The graphical abstract was created using BioRender (BioRender.com), and Figure A1C in Appendix A was generated using R Studio (R 4.3.3 GUI 1.80 Big Sur ARM build (8340)). These tools were used exclusively for data visualization and graphical presentation. They did not influence the study design, data analysis, or interpretation of the results.

## 5. Conclusions

The present study identified NP-related genes, developed predictive models, and established a link between NP and glucose metabolism. A gene signature comprising three specific genes was constructed, demonstrating its diagnostic potential. Additionally, we found that the expression of glucose metabolism-related genes was upregulated at the progression of the neuroinflammatory process in the PSNL model, providing new insights into the metabolic regulation of NP.

## Figures and Tables

**Figure 1 ijms-25-13503-f001:**
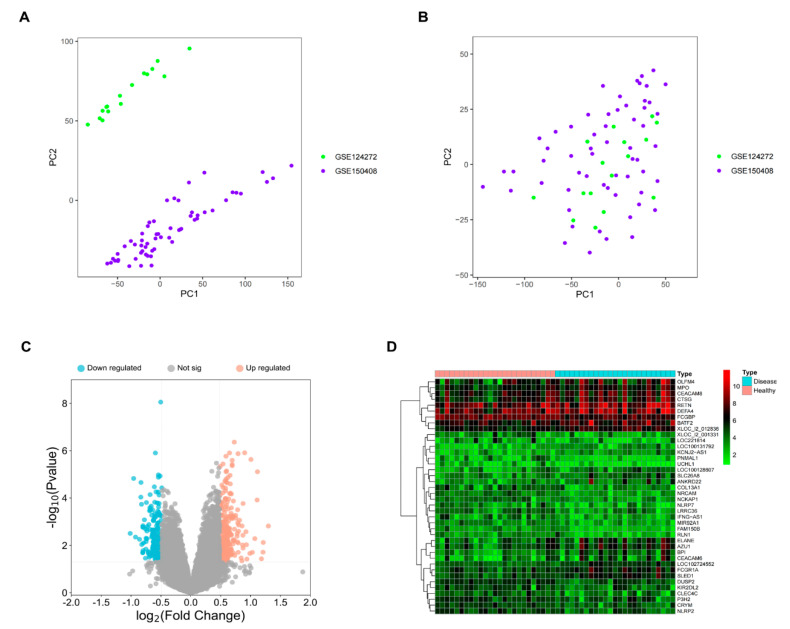
Identification of differentially expressed genes (DEGs) in neuropathic pain (NP) patients and healthy controls. (**A**) Principal component analysis (PCA) plot of the two datasets before batch effect correction, showing clear batch-related variability. (**B**) PCA plot of the two datasets after batch effect correction, demonstrating successful integration with reduced batch effects. (**C**) Volcano plot displaying all DEGs, with upregulated genes in red, downregulated genes in blue, and non-significant genes in gray. (**D**) Heatmap of the top 20 upregulated and downregulated DEGs, highlighting their differential expression between NP patients and healthy controls.

**Figure 2 ijms-25-13503-f002:**
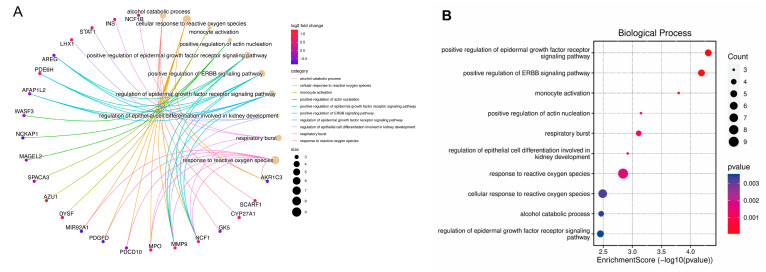
Gene Ontology (GO) and Kyoto Encyclopedia of Genes and Genomes (KEGG) pathway analyses of differentially expressed genes (DEGs). (**A**,**C**,**E**) Circular plots showing the top enriched GO terms for DEGs in the categories of biological process (BP) (**A**), cellular component (CC) (**C**), and molecular function (MF) (**E**). Each plot connects DEGs to their corresponding GO terms, with line colors reflecting the log fold change of the genes and circle sizes representing the number of associated genes for each term. (**B**,**D**,**F**) Dot plots summarizing the GO enrichment results for BP (**B**), CC (**D**), and MF (**F**). The x-axis indicates the enrichment score (−log10(*p*-value)), while dot colors represent *p*-value significance, and dot sizes correspond to the number of DEGs associated with each term. (**G**) Dot plot of KEGG pathway enrichment analysis for DEGs. Each dot represents a pathway, with the x-axis showing the enrichment score, dot color reflecting *p*-value significance, and dot size indicating the number of genes involved in each pathway.

**Figure 3 ijms-25-13503-f003:**
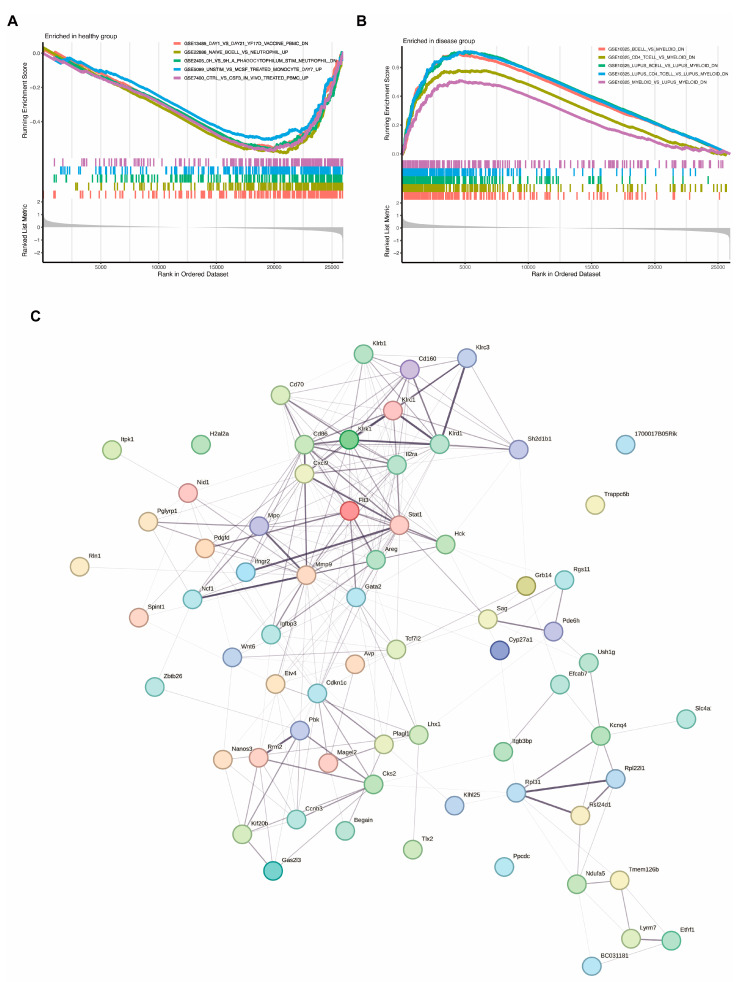
Gene set enrichment analysis (GSEA) and protein–protein interaction (PPI) network analysis. (**A**) GSEA plot of pathways enriched in the healthy group. The x-axis represents ranked genes, and the y-axis shows the enrichment score, highlighting pathways significantly upregulated in healthy individuals. (**B**) GSEA plot of pathways enriched in the diseased (IVDD) group. The plot illustrates pathways significantly upregulated in the diseased group, emphasizing key processes associated with disease progression. (**C**) PPI network of differentially expressed genes (DEGs). The network illustrates interactions among DEGs, with hub genes identified based on higher connectivity, indicating their potential regulatory roles in the associated pathways.

**Figure 4 ijms-25-13503-f004:**
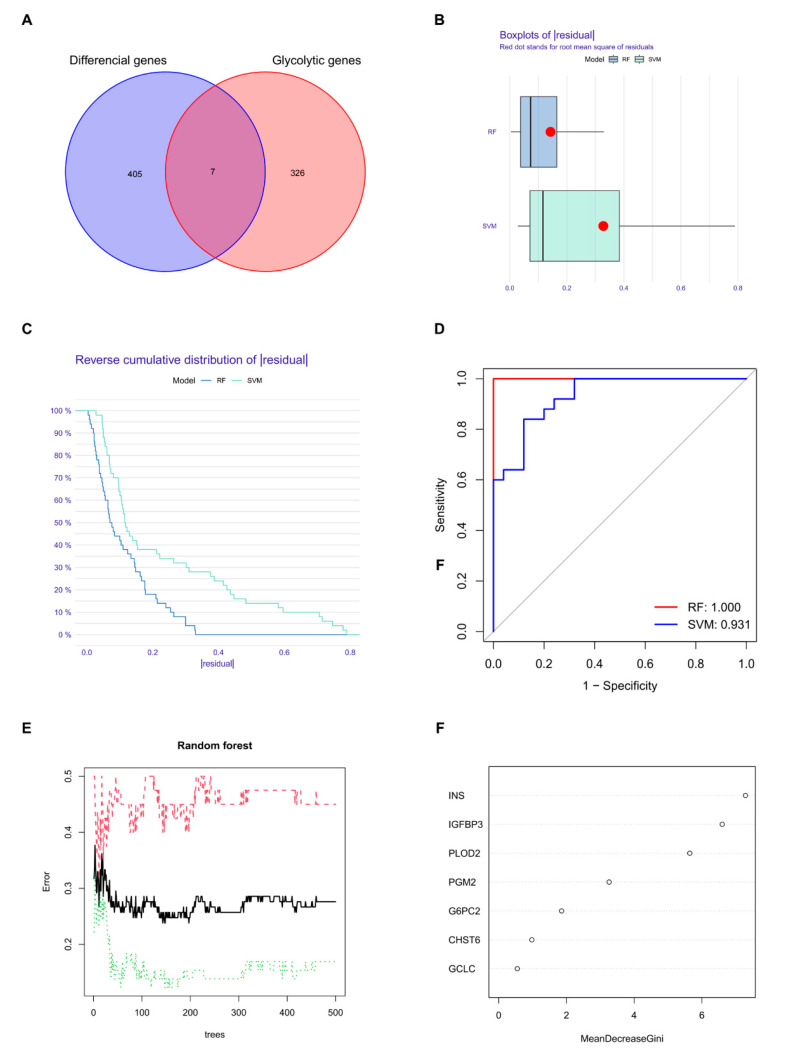
Identification of disease characteristic genes by machine learning model. (**A**) Venn diagram showing the intersection of differentially expressed genes (DEGs) with glucose-related genes, yielding seven candidate genes for further analysis. (**B**) Boxplots comparing residuals between the random forest (RF) and support vector machine (SVM) models, where lower residuals indicate better performance of the RF model. (**C**) Reverse cumulative distribution of residuals in the RF and SVM models, further demonstrating the lower residuals achieved by the RF model. (**D**) Receiver operating characteristic (ROC) curves of the RF and SVM models. The RF model achieved an AUC of 1.000, compared to 0.931 for the SVM model, suggesting higher diagnostic precision. (**E**) Importance scores of the seven candidate genes calculated by the RF model. The red dashed line represents the error rate for the disease group (IVDD), the green dotted line represents the error rate for the healthy group, and the black solid line represents the overall error rate. As the number of trees increases, the overall error rate and individual class error rates stabilize, indicating model convergence. (**F**) Expression patterns of the three key genes (*Ins*, *Igfbp3*, and *Plod2*) identified by the RF model, selected for follow-up analyses due to their strong performance.

**Figure 5 ijms-25-13503-f005:**
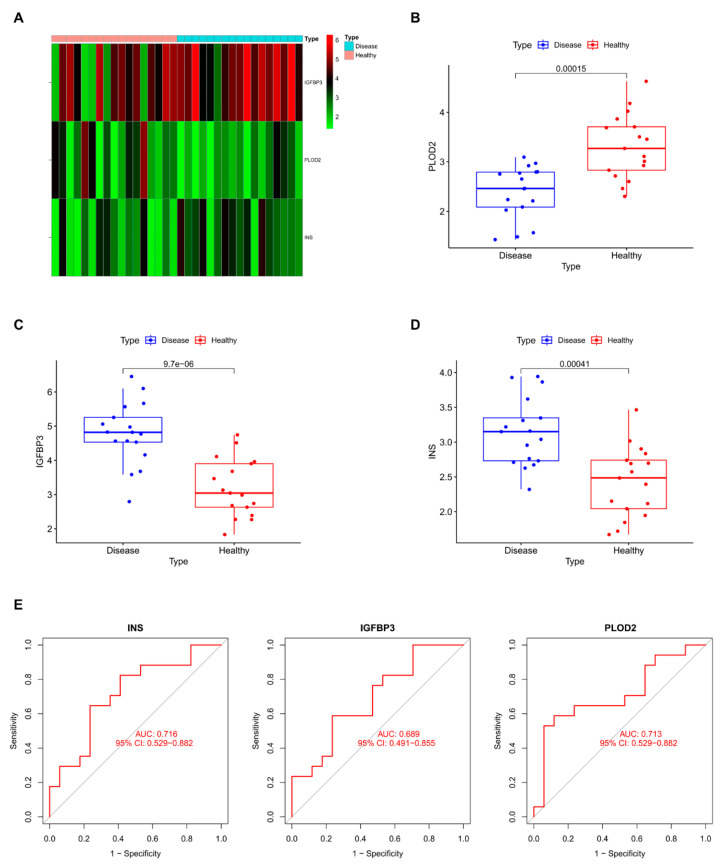
Expression analysis and diagnostic performance of disease characteristic genes. (**A**) Heatmap displaying the expression levels of the three characteristic genes (*Ins*, *Igfbp3*, and *Plod2*) in diseased (neuropathic pain) and healthy groups. Red and blue indicate higher and lower expression levels, respectively. (**B**) Box plot showing the expression level of *Plod2*, which is significantly lower in the diseased group compared to the healthy group. (**C**) Box plot showing the expression level of *Igfbp3*, which is significantly higher in the diseased group compared to the healthy group. (**D**) Box plot showing the expression level of *INS*, which is significantly higher in the diseased group compared to the healthy group. Statistical significance for differential expression is presented in each figure. (**E**) Receiver operating characteristic (ROC) curve illustrating the diagnostic performance of the three genes (*Ins*, *Igfbp3*, and *Plod2*) in IVDD. The area under the curve (AUC) values demonstrates the predictive accuracy of these genes in distinguishing disease from healthy groups.

**Figure 6 ijms-25-13503-f006:**
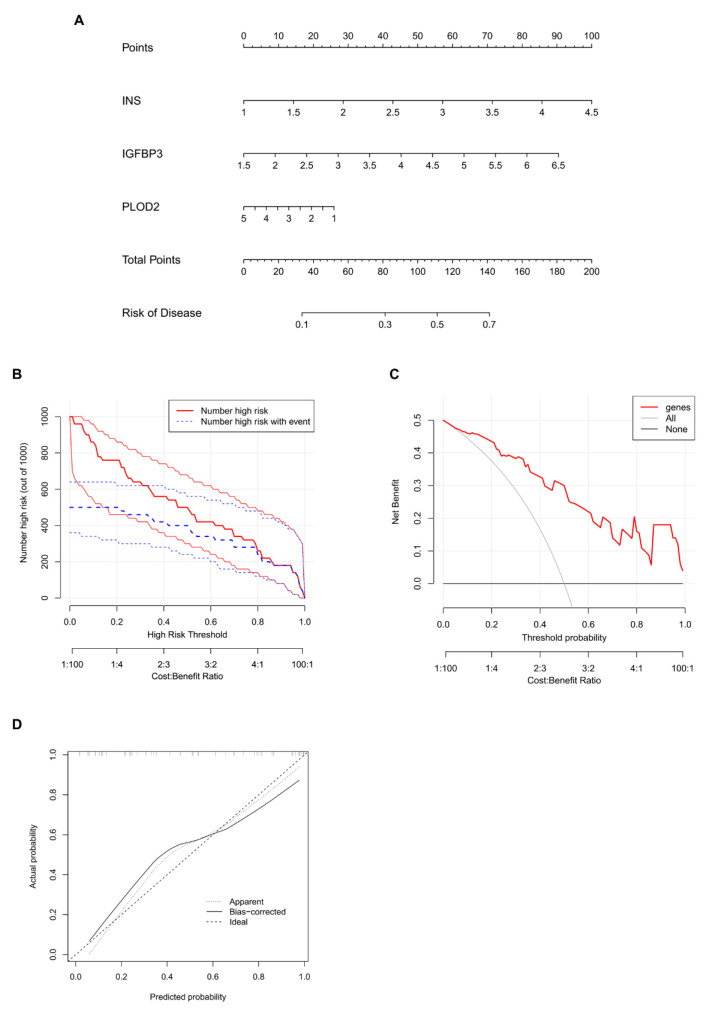
Construction of a nomogram. (**A**) Nomogram developed based on the characteristic genes (*Ins*, *Igfbp*3, and *Plod2*) to predict the likelihood of the disease. Each gene is assigned a score, and the total score corresponds to a predicted probability of IVDD. (**B**) Clinical impact curve illustrating the negligible difference between the predicted and actual positive rates, validating the reliability of the nomogram for clinical application. (**C**) Decision curve analysis (DCA) demonstrating the net benefit of the nomogram across a range of threshold probabilities, indicating its high clinical value. (**D**) Calibration curve comparing the predicted positive rate from the nomogram with the actual observed positive rate. The curve’s alignment with the diagonal line indicates the nomogram’s strong diagnostic accuracy for IVDD.

**Figure 7 ijms-25-13503-f007:**
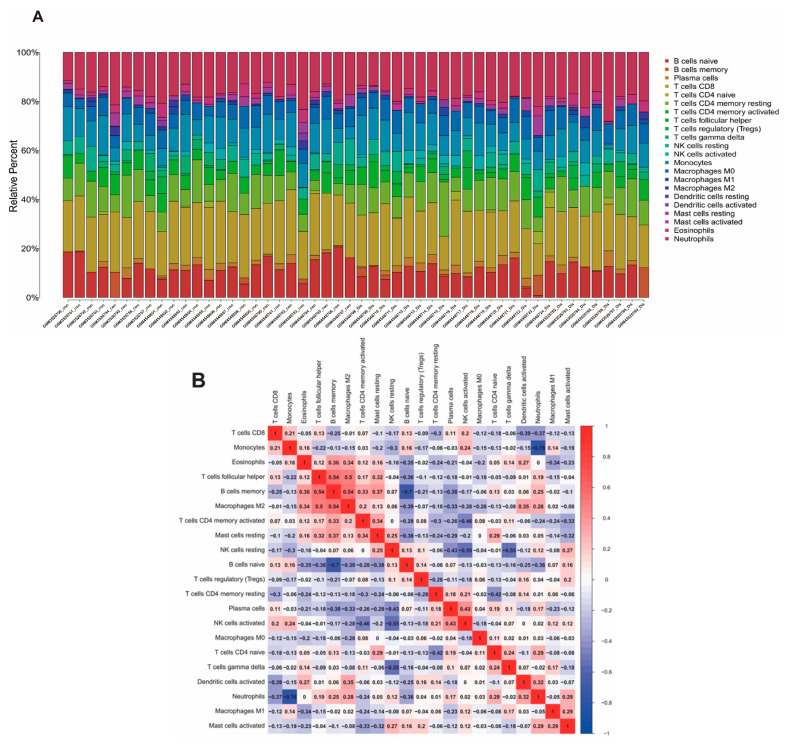
Immune infiltration analyses. (**A**) Heatmap displaying the relative percentages of 22 immune infiltrating cell types within the NP immune microenvironment. Each cell type’s proportion is calculated relative to the total immune cell population in individual samples, representing the composition of immune infiltration. (**B**) Correlation matrix showing the relationships between the 22 immune infiltrating cell types in the IVDD disease/NP group. Red and blue shades indicate positive and negative correlations, respectively. (**C**) Bar plot depicting the infiltration levels of immune cell types across individual samples from IVDD disease/NP patients, illustrating sample-specific immune variation. (**D**) Box plots comparing the levels of the 22 immune infiltrating cell types between diseased and healthy groups. Asterisks denote immune cell types that are differentially expressed between groups. (**E**) Correlation analysis between characteristic genes (*Igfbp3*, *Plod2*, and *Ins*) and the identified immune cell types. Positive correlations were observed between *Igfbp3*/*Plod2* and several immune cell types, while *Ins* expression was negatively associated with plasmacytoid dendritic cells and central memory CD8+ T cells.

**Figure 8 ijms-25-13503-f008:**
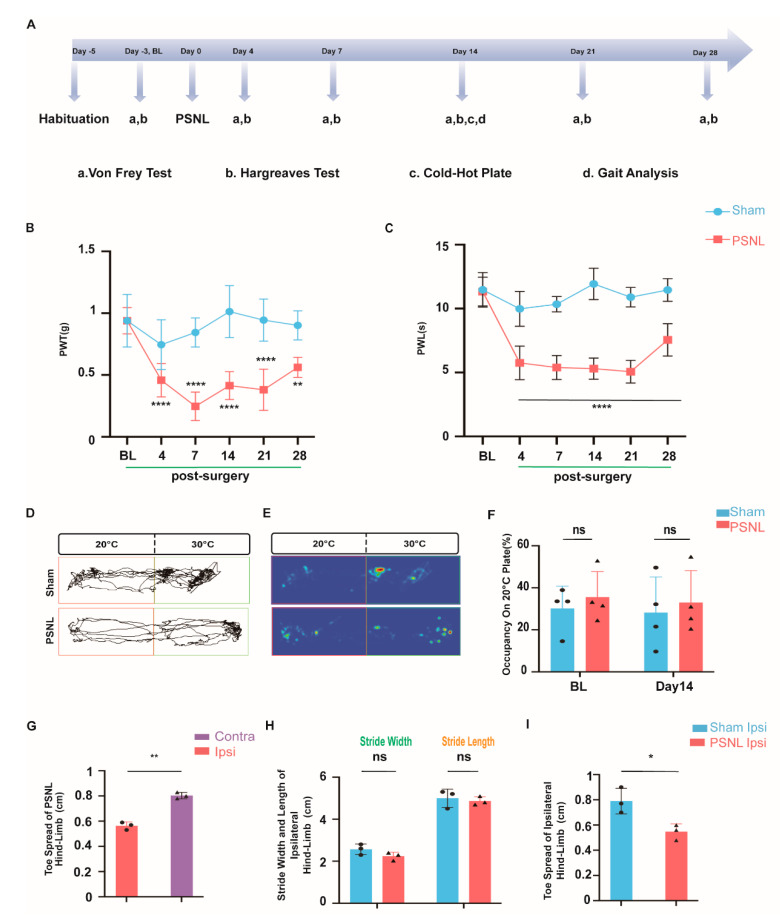
Behavioral test PSNL-induced hypernasality on day 14 post-surgery. (**A**) Schematic illustration of behavioral test design and timeline. (**B**) Time course of mechanical allodynia, shown as the von Frey force threshold for withdrawal, in sham and PSNL groups. (** *p* < 0.01; **** *p* < 0.0001 vs. sham; sham, n = 6; PSNL, n = 6). Two-way repeated-measures analysis of variance (ANOVA) followed by Sidak’s multiple comparison. (**C**) Time course of the thermal withdrawal latency (s) in sham or PSNL mice. (**** *p* < 0.0001 vs. sham; sham, n = 6; PSNL, n = 6). Two-way repeated-measures analysis of variance (ANOVA) followed by Sidak’s multiple comparison. (**D**) Representative track images and (**E**) Heatmap image of two-plate preference test at 30 °C versus 20 °C at 14 days post-operation. (**F**) Occupancy quantification of 30 °C versus 20 °C at baseline (left) and 14 days post-operation (right) (left: *p* = 0.5918, right: *p* = 0.6416; sham vs. PSNL; sham, n = 4; PSNL, n = 4). Unpaired *t*-test. (**G**) Changes in gait characteristics: toe spread of PSNL contralateral VS PSNL ipsilateral; (** *p* < 0.01 vs. PSNL contralateral; PSNL, n = 3). Unpaired *t*-test. (**H**) Changes in gait characteristics: stride length (left) and stride width (right) of sham ipsilateral VS PSNL ipsilateral; (Left: *p* = 0.3717, right: *p* = 0.8295; sham vs. PSNL; sham, n = 4; PSNL, n = 4). Unpaired *t*-test. Unpaired *t*-test. (**I**) Changes in gait characteristics: toe spread of sham ipsilateral VS PSNL ipsilateral; (* *p* < 0.05 vs. PSNL contralateral; PSNL, n = 3). Unpaired *t*-test.

**Figure 9 ijms-25-13503-f009:**
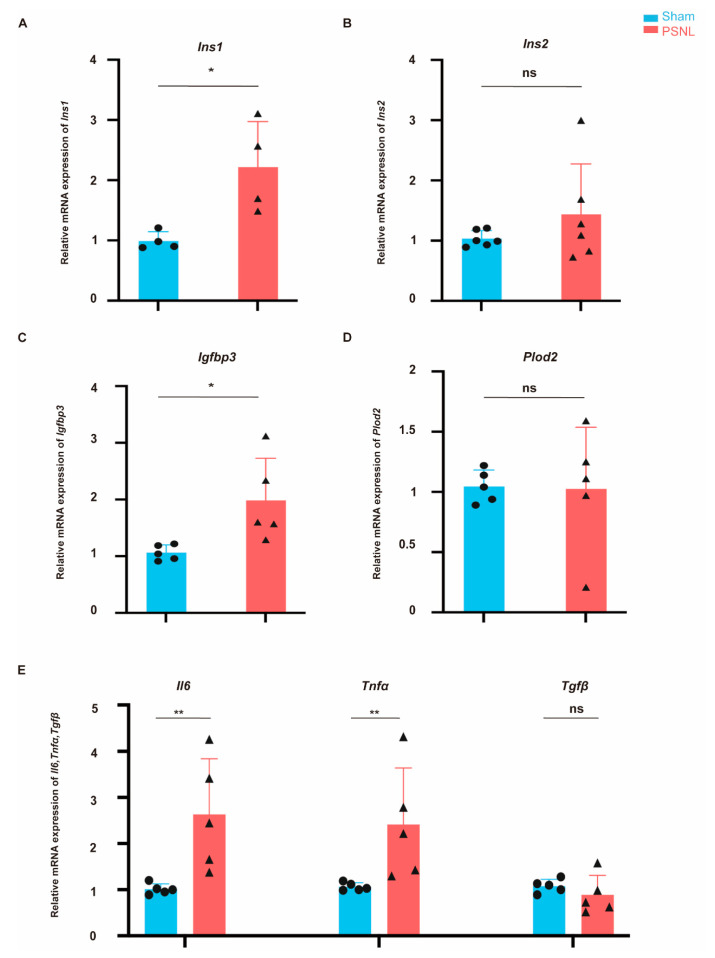
qRT-PCR showing mRNA expression levels of (**A**) *Ins1* (* *p* < 0.05 vs. sham); (**B**) *Ins2* (*p* = 0.6788); (**C**) *Igfbp3* (* *p* < 0.05 vs. sham); (**D**) *Plod2* (*p* = 0.6995) and (**E**) inflammatory related genes: *Il6* (** *p* < 0.01 vs. sham), *Tnfα* (** *p* < 0.01 vs. sham), and *Tgfβ* (*p* = 0.6836) in sham and PSNL at 14 days post-surgery. Gene expression was normalized to that of β-actin and is shown relative to sham, which is arbitrarily defined as 1 (sham vs. PSNL, n = 4~6/group). Unpaired *t*-test.

**Figure 10 ijms-25-13503-f010:**
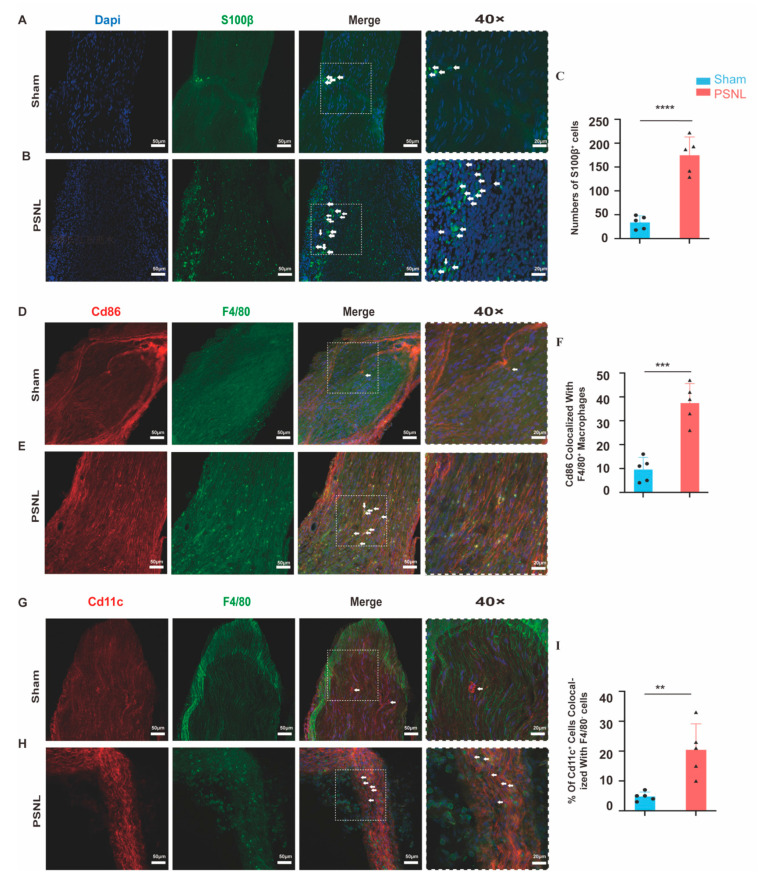
Representative images of immunofluorescence of S100β in (**A**) sham and in (**B**) PSNL at 14 days post-surgery. Scale bars, 50 μm. Dashed rectangles indicate higher magnification (40×); scale bars, 20 μm. Arrows indicate S100β-positive cells. (**C**) Quantification of the numbers of sciatic nerve marked by S100β positive cells in sham and in PSNL (**** *p* < 0.001 vs. sham; sham, n = 5; PSNL, n = 5). Unpaired *t*-test. Representative images of immunofluorescence of Cd86 and F4/80 in (**D**) sham and in (**E**) PSNL at 14 days post-surgery. Scale bars, 50 μm. Dashed rectangles indicate higher magnification (40×); scale bars, 20 μm. Arrows indicate Cd86 and F4/80 double-positive (co-staining) cells. (**F**) Quantification of the percentage of sciatic nerve marked by both Cd86 and F4/80 positive cells over total cells (Dapi) in sham and in PSNL (*** *p* < 0.001 vs. sham; sham, n = 5; PSNL, n = 5). Unpaired *t*-test. Representative images of immunofluorescence of Cd11c. and F4/80 in (**G**) sham and in (**H**) PSNL at 14 days post-surgery. Scale bars, 50 μm. Dashed rectangles indicate higher magnification (40×); scale bars, 20 μm. Arrows indicate Cd11c negative and F4/80 positive cells. (**I**) Quantification of the percentage of sciatic nerve marked by Cd11c negative and F4/80 positive cells over total cells (Dapi) in sham and in PSNL (** *p* < 0.01 vs. sham; sham, n = 5; PSNL, n = 5). Unpaired *t*-test.

**Table 1 ijms-25-13503-t001:** Primer sequences for target genes used in this study.

Genes	Forward/Reverse	Sequence 5′-3′
*Ins1*	Forward	GTCAAACAGCATCTTTGTGGTC
	Reverse	GGACTTGGGTGTGTAGAAGAAG
*Ins2*	Forward	AGCAGCACCTTTGTGGTT
	Reverse	CTCCAGTTGTGCCACTTGT
*Igfb3*	Forward	AACCTGCTCCAGGAAACATC
	Reverse	GGAACTTGGAATCGGTCACT
*Plod2*	Forward	GCGCATCCCTGCAGATAAAT
	Reverse	GACCTTGACCAAGAACCTTCAC
*Hif1α*	Forward	ATAGCTTCGCAGAATGCTCAGA
	Reverse	CAGTCACCTGGTTGCTGCAA
*Il6*	Forward	GAAACCGCTATGAAGTTCCTCTCTG
	Reverse	TGTTGGGAGTGGTATCCTCTGTGA
*Tnfα*	Forward	GGGTGTTCATCCATTCTC
	Reverse	GGAAAGCCCATTTGAGT
*Tgfβ*	Forward	CGAAGCGGACTACTATGCTAAA
	Reverse	CTGTATTCCGTCTCCTTGGTTC
*β-actin*	Forward	CCTAGACTTCGAGCAAGAGA
	Reverse	GGAAGGAAGGCTGGAAGA

**Abbreviation of genes**: *Ins1*: insulin 1; *Ins2:* insulin 2; *Igfb3*: insulin-like growth factor binding protein 3; *Plod2*: procollagen lysine, 2-oxoglutarate 5-dioxygenase 2; *Hif1α*: hypoxia-inducible factor 1, alpha subunit; *Il6*: interleukin 6; *Tnfα*: tumor necrosis factor alpha; *Tgfβ*: transforming growth factor beta; *β-actin*: actin, beta.

## Data Availability

The data presented in this study are openly available in the Gene Expression Omnibus (GEO) database at accession numbers GSE124272 and GSE150408. The datasets can be accessed at https://www.ncbi.nlm.nih.gov/geo/. Further inquiries can be directed to the corresponding author.

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
