# Peer review of "Discovery of Glucose Metabolism-Associated Genes in Neuropathic Pain: Insights from Bioinformatics"

_ijms, 2024, doi:10.3390/ijms252413503_

Round 1
Reviewer 1 Report
Comments and Suggestions for Authors
Yu et al. present an adequate and highly novel original manuscript. The manuscript has great potential and I must congratulate the authors. The suggestions are minor:
-The title should have more impact. I suggest that the authors improve this point.
-The abstract of the manuscript should be expanded.
-The authors should improve the figure legends and make the figures more self-explanatory.
-Figure 3 should be improved and better described.
-Figure 7 should be divided into several, it is not visible well.
-The immunofluorescence histology images should be better described and include higher magnifications.
-The authors should include a graphical abstract.
Author Response
-The authors should improve the figure legends and make the figures more self-explanatory.
We agreed with your professional idea and we changed all the figure legend to make it clear and reasonable (hopefully).
-Figure 3 should be improved and better described.
As mentioned in the last reply we tried to described figure three in a more scientific way.
-Figure 7 should be divided into several, it is not visible well.
Thank you for your warm response,we adjust the figure quality and presented it devidely.
-The immunofluorescence histology images should be better described and include higher magnifications.
I would like to seek clarification regarding one specific request from a reviewer, who suggested including higher magnification images in the immunofluorescence analysis. Currently, the manuscript already provides 40x magnification images (LZW format). Based on my expertise, I believe using 60x magnification may compromise the overall presentation of the images, potentially making it difficult to interpret key features. I would appreciate your guidance on whether this adjustment is necessary.
-The authors should include a graphical abstract.
We appreciate your helpful suggestion and drew a graphic abstract.
Reviewer 2 Report
Comments and Suggestions for Authors
The text contains comments on “Identification and validation of potential glucose metabolism-related genes in neuropathic pain based on bioinformatics analysis”
To my opinion, the manuscript is well written on sufficiently good English. The experimental design is logically structured and the results fully explain the obtained data in align with the aim of the study.
I have listed some minor corections:
In Introduction section, could you please also discus if your study is performed for first type? What is the novelty in it, what gaps in the field it will fill and what are the benefits of it compared to other studies if applicable.
I think that this discussion should concern also the conclusion section.
Author Response
In Introduction section, could you please also discus if your study is performed for first type?
Thank you for your comment and we presented the novelty in the introduction.
What is the novelty in it, what gaps in the field it will fill and what are the benefits of it compared to other studies if applicable.
I think that this discussion should concern also the conclusion section.
Thank you for your professional suggestion and we compared the presented study to the others and emphasize the gaps and benefits in the revised version.
Reviewer 3 Report
Comments and Suggestions for Authors
Dear authors,
the presented article is of great scientific importance for the identification of new biomarkers for neuropathic pain resulting from glucose metabolism disorders.
I appreciated the way in which the hypothesis was formulated and the way in which this research was conducted.
As small recommendations, I would suggest to the authors that the mechanisms presented in the introduction be accompanied by an image that would allow for easier tracking of them.
Also, as a non-specialist, I would like the figures to be better explained (legends), making them understandable on their own.
The authors state that the research has certain flaws. Can the authors propose methods to resolve them?
My recommendation is to publish with minor revisions after resolving the observations made.
Author Response
As small recommendations, I would suggest to the authors that the mechanisms presented in the introduction be accompanied by an image that would allow for easier tracking of them.
We appreciate your helpful suggestion and drew a graphic abstract to make the idea more straight forward.
Also, as a non-specialist, I would like the figures to be better explained (legends), making them understandable on their own.
We agreed with your professional idea and we changed all the figure legend to make it clear and reasonable (hopefully).
The authors state that the research has certain flaws. Can the authors propose methods to resolve them?
Thank you for pointing this out and we had our understandings on flaw resolving in the last paragraph.